# Development of SSR Markers Linked to Stress Responsive Genes along Tomato Chromosome 3 (*Solanum lycopersicum* L.)

**DOI:** 10.3390/biotech11030034

**Published:** 2022-08-16

**Authors:** Mohammad Brake, Lana Al-Qadumii, Hassan Hamasha, Hussein Migdadi, Abi Awad, Nizar Haddad, Monther T. Sadder

**Affiliations:** 1Science Department, Jerash University, Jerash 26150, Jordan; 2Faculty of Science, Philadelphia University, Jerash 19392, Jordan; 3National Agricultural Research Center, Amman 19381, Jordan; 4Food Testing Lab, Jordan Standards and Metrology Organization, Amman 11194, Jordan; 5Plant Biotechnology Lab, Department of Horticulture and Crop Science, School of Agriculture, University of Jordan, Amman 11942, Jordan

**Keywords:** BACs, heterozygosity, polymorphic information content, SSRs, stress, tomato

## Abstract

This study aimed to develop novel SSR markers in tomato. Several BAC clones along chromosome 3 in tomato were selected based on their content. The criteria was the availability of genes, either directly or indirectly related to stress response (drought, salinity, and heat) in tomato. A total of 20 novel in silico SSR markers were developed and 96 important nearby genes were identified. The identified nearby genes represent different tomato genes involved in plant growth and development and biotic and abiotic stress tolerance. The developed SSR markers were assessed using tomato landraces. A total of 29 determinate and semi-determinate local tomato landraces collected from diverse environments were utilized. A total of 33 alleles with mean of 1.65 alleles per locus were scored, showing 100% polymorphic patterns, with a mean of 0.18 polymorphism information content (PIC) values. The mean of observed and expected heterozygosity were 0.19 and 0.24, respectively. The mean value of the Jaccard similarity index was used for clustering the landraces. The developed microsatellite markers showed potential to assess genetic variability among tomato landraces. The genetic distance information reported in this study can be used by breeders in future genetic improvement of tomato for tolerance against diverse stresses.

## 1. Introduction

Tomato (*Solanum lycopersicum* L.) is one of the most important and widespread vegetable crops around the world. It is the second most consumed vegetable after potato with a total production reaching 187 million tones [1]. Tomato belongs to the family Solanaceae, which includes several agronomic importance crops, such as potato and pepper. Tomato is a diploid (2n = 2x = 24) plant, and its genome is approximately 950 Mb in size [2]. The breeding system in tomato vary from allogamous to facultative allogamous to autogamous [3].

Both biotic and abiotic stresses are important constraints to tomato productivity. These stresses affect plant growth during all developmental stages when imposed either individually or combined [4,5,6,7,8,9,10]. Climate changes, including drought, salinity, and heat, are the most important factors that reduce agricultural crop yields in arid and semi-arid regions, which threaten the level of food security.

There are more than 7500 tomato landraces and varieties successfully bred and grown for various purposes worldwide [11]. These tomato genetic resources have special importance in breeding programs as sources of desired genes for different purposes, such as disease resistance and agronomic traits [12,13], and they serve as a model organism for genetic and developmental studies [14,15]. Study of phenotypic and genetic diversity in tomato landrace collections is important for germplasm preservation, exploitation, and utilization of these genetic resources. In addition, the characterization of highly diversified materials with molecular markers offers a unique opportunity to define significant marker-trait associations with biological and agronomic interest. To this end, various marker techniques have been successfully applied, either individually or in combination, to study the genetic diversity of various plant species [16]. Unfortunately, the gradual disappearance of many tomato landraces in favor of high-yielding cultivars is likely to erode the genetic base of tomato [17].

Simple sequence repeats (SSRs) is an important source of DNA markers due to their high reproducibility, multiallelic nature, co-dominant inheritance, abundance, and wide genome coverage. Development of SSR markers from map-referenced BAC clones was a very effective means of targeting markers to marker scarce-positions in the genome [18]. In silico mining of SSRs from sequence databases [19] provides an attractive alternative to the molecular approaches. Not only is the in silico approach time and cost effective but it also allows for the discovery of SSRs from expressed sequence tags (ESTs) that represent the coding region of genome. SSR markers have been successfully utilized to analyze genetic diversity in tomato [20,21,22,23]. SSR marker are very crucial in breeding and analysis of plant abiotic stresses [24].

We initiated this study to develop, validate, and map new SSR markers based on in silico analysis, which are tightly linked to putative response genes related to biotic and abiotic stresses using tomato landraces.

## 2. Materials and Methods

### 2.1. Plant Materials

Seeds from a total of 29 tomato (*Solanum lycopersicum* L.) landraces were obtained from the National Agricultural Research Center (NARC), Jordan, where the tomato seeds collected from local farmers throughout the country are kept in the medium-term germplasm seed gene bank. These landraces were collected from different geographical origins [25]. Table 1 shows some of the vegetative characters of the landraces included in the study. Seeds were planted in a growth room for 3 weeks and young leaves were collected and stored under −20 °C for genomic DNA extraction.

### 2.2. DNA Extraction

Genomic DNA was extracted using Wizard genomic DNA purification kit (Promega, Madison, WI, USA) and was used according to instructions provided by the manufacturer. Then, DNA was stored at −20 °C. Genomic DNA was electrophoresed at 0.8% agarose gel containing ethidium bromide and detected under UV-light. DNA concentration was determined using spectrophotometry.

### 2.3. SSR Marker Development

Mainly, tomato chromosome three was selected because it showed potential responsive biomarkers for both biotic and abiotic stresses [7,9]. Nonetheless, some SSR markers were found on other chromosomes (1, 1 and 2 on chromosomes 4, 10, and 12, respectively). Bacterial artificial chromosome (BAC) clones along chromosome 3 were retrieved from Sol Genomics Network (SGN) (https://solgenomics.net/) (accessed on 1 June 2022) and corresponding Genbank accession numbers were determined (NCBI, https://www.ncbi.nlm.nih.gov/) (accessed on 1 June 2022). Whole BAC sequences were searched for stress related genes. Concurrently, the same BACs were screened for available SSR sequences using the Simple Sequence Repeat Identification Tool (SSRIT) [19]. Adjacent flanking sequences for SSR loci were selected and then used to develop SSR specific PCR primers (Table 2). Detected SSR markers were viewed with Jbrowse available in SGN (2022), and adjacent stress responsive genes were retrieved and tabulated.

PCR reactions were performed in a 10 μL volume consisting of 20 ng of DNA, 1.0 unit of DNA Taq Polymerase (Promega), 1 × PCR buffer (Promega), 1.5 mM MgCl_2_, 0.2 mM of each dNTP (Promega), 0.5 μM of tailed forward primer (Integrated DNA technology, Coralville, IA, USA), 0.03 μM tailed labeled with IRD700 (Integrated DNA technology), and 1.5 μM of reverse primer (Integrated DNA technology). The forward primer was “tailed” by the inclusion of 19 extra nucleotides at the 5′ end, which facilitated the labeling of the products. The reactions were carried out in a thermo cycler Perkin-Elmer 9700 (Applied Biosystems, Waltham, MA, USA), with the following profile: 95 °C for 5 min, 20 cycles at 95 °C for 20 s, annealing temperature (65 °C) for 30 s, decreasing 0.5 °C/cycle, extension temperature 72 °C for 30 s; followed by 20 cycles at 95 °C for 30 s, annealing temperature 55 °C for 30 s, and 72 °C for 30 s with a final extension at 72 °C for 10 min. SSR markers were profiled using a LI-COR Bioscience 4300 DNA Analyzer, 1 μL of the product was loaded onto a 6% polyacrylamide gel after mixing with 0.5 μL stop solution (Li-COR), and electrophoresed at 1500 Volts.

### 2.4. Data Analysis

The Jaccard similarity matrix was used for cluster analysis, using the unweighted pair group method arithmetic average to study the genetic relationships among the cultivars [26]. These coefficients were used to construct dendrogram, using the unweighted pair group method with arithmetic average (UPGMA); the robustness of internodes was assessed by bootstrap analysis with 1000 replicates and principal coordinate analysis (PCoA) was performed using the PAST program [27]. For each primer pair, Microsatellite-Toolkit for Excel [28] was used for estimating mean number of alleles, observed and expected heterozygosities (Ho, He) [29], polymorphism information content (PIC) according to [30], power of discrimination was calculated with the formula PD = 1 − Σgi2, where gi is the frequency of the cultivar at locus I [31].

## 3. Results

A total of 20 novel microsatellites were developed in silico. The microsatellites are distributed among chromosomes 3, 4, 10, and 12 (Figure 1). In total, 16 loci were located on chromosome 3, while 1, 1, and 2 loci were located in chromosomes 4, 10, and 12, respectively. All original BAC clones were assigned to chromosome 3 in tomato (when this study was carried out). However, new data curation remapped several clones to other chromosomes. For example, 2 loci (ju022 and ju023) were found to be located on chromosome 12 based on the clone C12HBa0270F24, which was originally mapped on chromosome 3 based on the clone number C03HBa0270F24. In addition, the ju006 marker found in the original clones (number C03HBa0224P23) was assigned to chromosome number 3 but found recently to be located on chromosome 10. Nonetheless, 4 loci are clustered at the telomeric end of the short arm of chromosome 3, while 10 loci are clustered at the telomeric end of the long arm of the same chromosome, similarly, 2 other loci were found in the telomeric region of the short arm of chromosome 12.

However, ju041 was located inside chromosome 3 as well as ju008. Likewise, ju039 and ju006 are also located toward the middle of chromosomes 4 and 10, respectively.

The chromosomal coordinates for the 20 novel developed microsatellite were determined using the Sol Genomics network (Appendix A). Ninety six nearby genes for the novel developed microsatellite were identified. The identified nearby genes representing different tomato genes, for example, loci ju006, ju014, ju015, ju017, and ju035 are located near Serine/Threonine protein kinase-related gene and loci ju010 and ju035 are located near MYB transcription factor gene, whereas ju014 is located near AP2-like ethylene-responsive transcription factor (Appendix A).

Genetic diversity was examined in Jordanian tomato landraces to validate novel SSR markers. In total, 33 polymorphic alleles with a 100 polymorphism percentage was achieved. The mean number of alleles per locus was 1.65. The highest value for observed heterozygosity was 0.81 recorded for ju007, while the lowest was 0.0 with an average of 0.19. Expected heterozygosity per locus ranged from 0.00 to 0.54, with an average of 0.24. The average value of PIC for the primer sets was 0.18, ranging from (0.0) to (0.38). PD varied from 0.32 for ju003 to 0.96 for ju023 and ju026, with an average of 0.68 (Table 3).

Pair-wise similarity values ranged from 0.00 to 0.89 and the overall accessions similarity showed an average of 0.47. The maximum similarity index (0.89) was recorded between accessions of 951AL’AL and 960SHATANA, while low values of genetic similarity between 961AINJNA and 988BAIDA, 989BAIDA, and 958SAKIB were reported (Figure 2).

The UPGMA cluster analysis of the accessions based on SSR data exhibited moderate clustering relationships, except for 951AL’AL and 960SHATANA (bootstrapping value 81%), two accessions from Ain AL-Baida (985BAIDA and 987BAIDA) with bootstrapping value of 65%, 952AL’AL and 995WMUSA with bootstrapping value of 58%, and the two accessions 964RHABA and 974bRHABA with 55% of bootstrapping value. At 50% of similarity value, two major groups were formed (Figure 3). Subclusters were revealed for major branches separating accessions from the same geographical distribution. Cluster 1 contained four accessions of Rhaba region and cluster 2 was further subdivided to subgroups compassed 19 accessions. In one subgroup two accessions from Rhaba (972RHABA and 973RHABA) grouped corresponding with their geographical area. Except 995WMUSA, which is cultivated in the southern part of Jordan, five accessions were grouped in a second subgroup, including two accessions from Al’al (951AL’AL and 952AL’AL), 975RHABA, 960SHATNA, and 956HEBRS, all were cultivated in the northern part of Jordan. Two accessions from the northern part of Jordan, which were 963RHABA and 996RHABA, and two from the south, 985BAIDA and 987BAIDA, were grouped in a third subgroup. A fourth subgroup contained three accessions from the southern part (983ABELand 984ABEL and 994aSHOBK) and two from the northern part (964RHABA and 974bRHABA). 955QSFA and 980aAFRA representing two diverse region formed a fifth subgroup. The remaining six accessions failed to form clusters and were individually separated.

Principal coordinate analysis (PCoA) was performed to validate genetic relationships among 29 accessions (Figure 4). The first two axes explained 37.9% of the variation, where the first coordinate accounted for 21.1% variation, while the second axis explained a 16.8% variation. Following the same trend of the dendrogram, moderate relationships and no specific geographic relationships were obtained. However, PCoA showed that all samples were distributed to the four parts of the coordinates with no specific aggregations.

Of all developed SSR markers, di-nucleotide were the most abundant with 12 loci, of these 4 loci are of AT repeat motif, followed by tri-nucleotide with 7 loci and tetra-nucleotide with 1 locus (Table 2).

## 4. Discussion

A total of 14 out of 20 novel developed microsatellites are clustered in telomeric and subtelomeric regions of chromosome 3. Furthermore, two other SSR loci were found located at sub-telomeric region of chromosome 12. The result is inconsistent with results, emphasizing that plant genes show clustering in telomeric and subtelomeric regions. Ref. [32] reported that the avenacin cluster (12 genes) lies in a subtelomeric region at the end of the long arm of chromosome 1.

Important genes for plant growth and development, disease resistance, and abiotic stress tolerance along with many other crucial processes in plants were found. For example, MYB transcription factor, which is located near loci ju010 and ju035, plays a key role in plant development, secondary metabolism, hormone signal transduction, disease resistance, and abiotic stress tolerance, [33,34,35], while calmodulin-binding heat shock protein, ap2-like ethylene-responsive transcription factor, and WRKY transcription factor, which are located near locus ju027, locus ju014, and the locus ju023, respectively, are involved in plant responses to abiotic stresses [36,37,38]. On the other hand, leucine-rich repeat family protein, which is located near locus ju007, provide recognition of pathogen products of avirulence (AVR) genes [39]. Other loci are located near other important genes (Table 4).

The high percentage of polymorphism (100%) obtained in the study was similar with other studies conducted in tomato landraces. Ref. [45] obtained 100% of polymorphism for 4 SSR primers out of 5 for 39 Jordanian tomato landraces. Ref. [46] reported 100% polymorphism using 9 tomato landraces form Jordan along with 1 commercial cultivar using SSR markers and [25] obtained 60% of polymorphism for the same landraces used in this study using ISSR markers. A high percentage of polymorphism for tomato landraces using different molecular markers from different countries was obtained from Saudi Arabia [47,48], Turkey, and Iran [49].

The observed distribution of repeat motif in SSR markers developed: 60% for di-nucleotide, 35% for tri-nucleotide, and 5% for tetra-nucleotide was in accordance with other studies about the nature of repeat motif in SSR markers in plant genomes. Ref. [50] reported that SSRs existed primarily as dinucleotide repeats and trinucleotide repeats, accounting for 97.59% of all SSRs. Dinucleotide repeats (74.56%) were the most abundant repeat unit, followed by tr- (23.08%) and tetra-(2.04%) in the Camellia japonica genome. Furthermore, out of the 15,498 SSR markers analyzed in the Platostoma palustre genome [51], 71.96%, 26.26%, and 1.52% was SSR with di-nucleotide, tri-nucleotide, and tetra-nucleotide repeat motif, respectively.

A strong correlation between genetic similarity values and geographical distribution were recorded between Jordanian tomato landraces included in the study. Our results were in agreement with other results for genetic diversity studies using tomato landraces from different parts of the world. For example, Ref. [20] proved that 14 florescent SSR markers were able to separate 15 local tomato landraces from Campania region (Southern Italy) according to their geographical distribution, and the UPGMA dendrogram supported by principal coordinate analysis (PCoA) revealed clusters of Saudi tomato landraces according to their geographical origin using SDS-PAGE and sequence-related amplified polymorphism (SRAP) markers [47].

Although, the dendrogram and the principle coordinate analysis shows a moderate relationship between landraces grouping and geographical region, some evidences of correlations between landraces and geographical region was observed. In the two main subclusters, which comprised 23 landraces out of 29 formed at 50% of genetic similarity, many landraces from the same geographical region are clustered together similar to the landraces in subcluster 1, which are from RHABA region and 5 landraces in the second subgroup of subcluster II. The obtained results could be supported by other results conducted by [20,25,47], using different tomato landraces and different molecular markers. In cases where landraces grouped according to geographical region, it could be explained by a reduced admixture of the gene pool between local farmers.

## 5. Conclusions

In this study, we found that most of the developed SSR markers are located in the telomeric region of both short and long arms of chromosome 3 and 12. Many important genes are identified near the developed SSR markers, a major group of these nearby genes are important responsive factors for abiotic tolerance and biotic resistance, whereas other genes are important for plant growth and development. The newly developed SSR markers were validated using a collection of Jordanian tomato landraces comprised of 29 landraces from different geographical areas of the country. A high percentage of polymorphism were found for all alleles. Some landraces showed most of the developed SSR markers, while others showed some of these SSR markers. In this regard, potential landraces were further selected for salinity stress analysis, using DGE of salinity responsive genes, to be used in our breeding program.

## Figures and Tables

**Figure 1 biotech-11-00034-f001:**
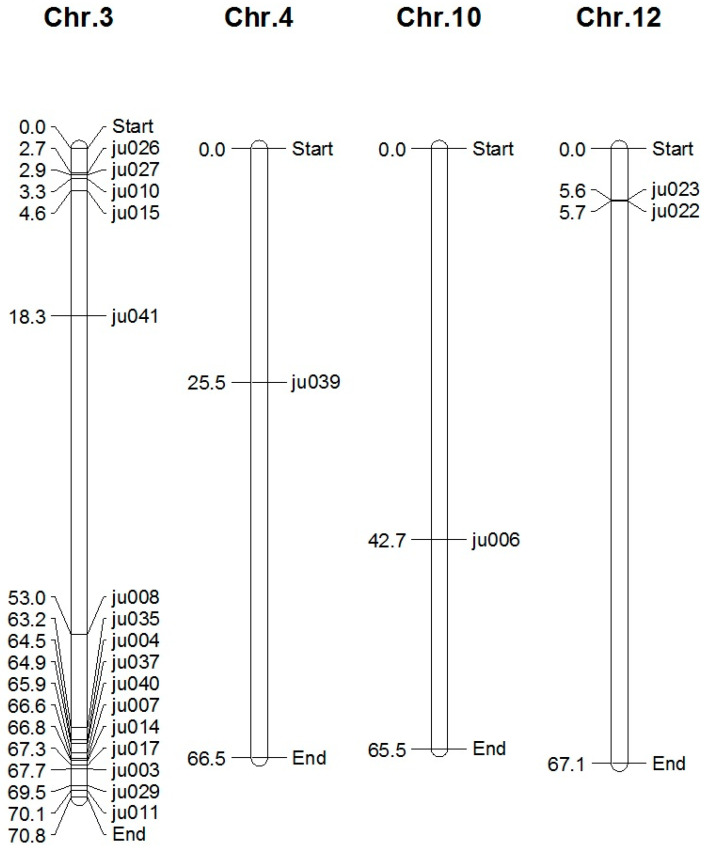
Distribution of novel microsatellite loci in different tomato chromosomes. Loci (right to chromosome) and their corresponding distance in Mb (left to chromosome).

**Figure 2 biotech-11-00034-f002:**
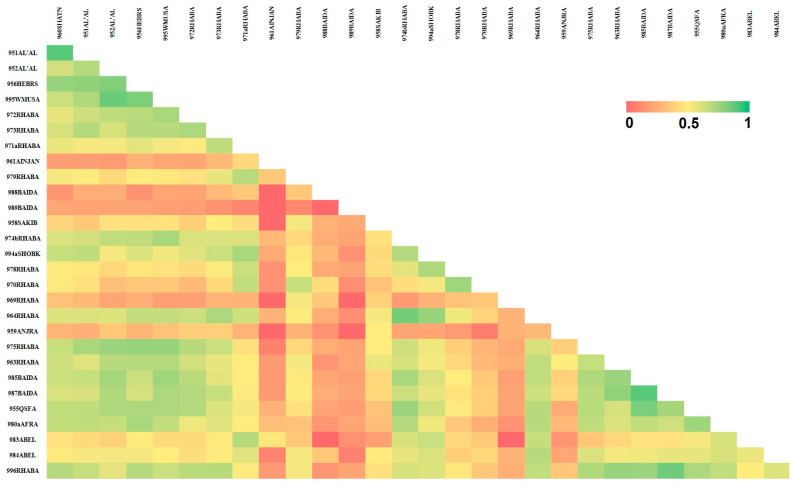
Jaccard’s similarity index (0–1) represented as a heatmap generated from developed SSR marker data for a collection of 29 tomato landraces (depicted as accession number and collection site).

**Figure 3 biotech-11-00034-f003:**
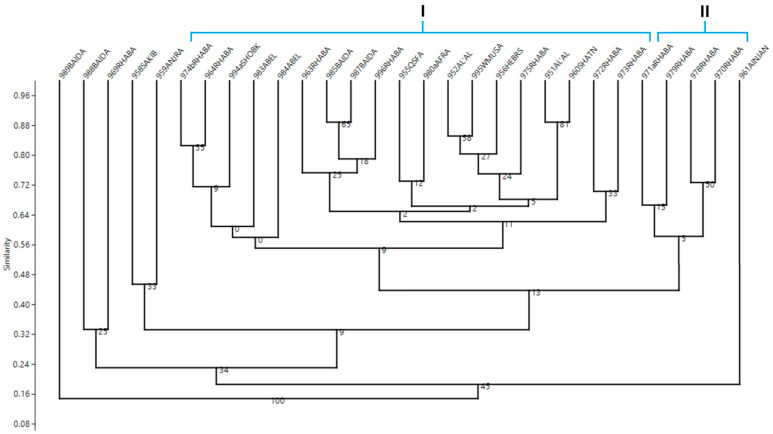
Dendrograms generated using unweighted pair group method with arithmetic average (UPGMA) analysis, showing relationships between 29 tomato landraces (Accession number and collection site), using SSR data based on Jaccard genetic similarity index. Bootstrap support value is given above each branch.

**Figure 4 biotech-11-00034-f004:**
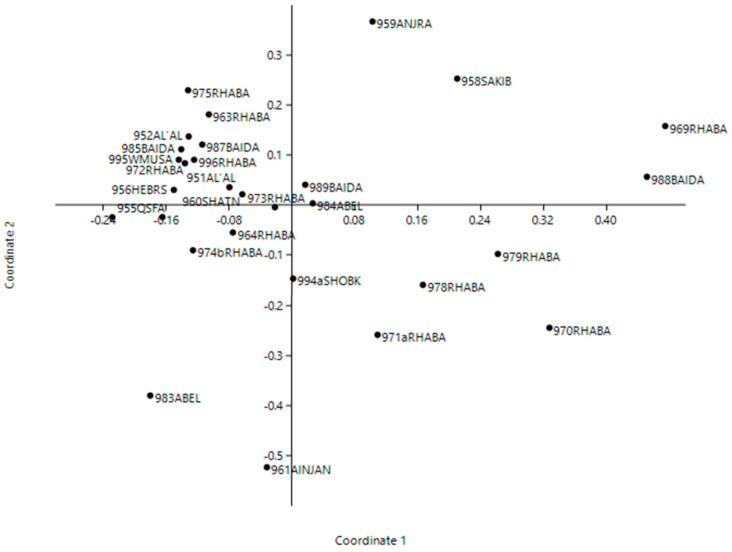
Principal coordinate analysis (PCoA) of 29 Jordanian tomato landraces (Accession number and collection site) based on 20 microsatellite loci. The first and second coordinate accounting for 21.1% and 16.8% variation, respectively.

**Table 1 biotech-11-00034-t001:** Vegetative characters of the 29 Jordanian tomato landraces.

Accession Number	Growth Type	Plant Size	Foliage Density	Growth Habit
951	Determinate	Large	Dense	Erect
952	Determinate	Med.–large	Dense	Prostrate branched
955	Determinate	Med.–large	Intermediate	Prostrate branched
956	Determinate	Small	Intermediate	Erect
958	Determinate	Medium	Intermediate	Erect less branched
959	Determinate	Med.–large	Intermediate	Half erect branched
960	Determinate	Small	Sparse	Erect less branched
961	Determinate	Large	Dense	Prostrate
963	Determinate	Large	Dense	Erect branched
964	Semi-determinate	Large	Dense	Erect
969	Determinate	Medium	Intermediate	Prostrate branched
970	Determinate	Med.–large	Dense	Erect branched
971A	Semi-determinate	Medium	Intermediate	Erect
971B	Determinate	Small-med.	Sparse	Prostrate branched
972	Determinate	Large	Dense	Erect branched
973	Determinate	Medium	Intermediate	Prostrate
975	Determinate	Medium	Sparse	Erect branched
978	Semi-determinate	Large	Dense	Prostrate
979	Determinate	Med.–large	Intermediate	Prostrate
980A	Determinate	Medium	Intermediate	Erect branched
983	Determinate	Small–med.	Intermediate	Prostrate branched
984	Determinate	Med.–large	Intermediate	Prostrate branched
985	Determinate	Med.–large	Sparse	Erect branched
987	Determinate	Large	Dense	Prostrate branched
988	Determinate	Medium	Dense	Less erect
989	Semi-determinate	Large	Dense	Half erect branched
994A	Determinate	Large	Dense	Prostrate
995	Determinate	Small–med.	Intermediate	Prostrate branched
996	Determinate	Large	Dense	Erect branched

**Table 2 biotech-11-00034-t002:** The 20 novel microsatellite loci developed in the study along with their related PCR information.

BAC Accession	Clone Name	SSR Locus	Chr	Primer Sequence (5’–3’)	Repeat Motif	Allele Size (s) (bp)	Tm (°C)
AC235792	C03HBa0029M12	ju003	3	F-ATGGTGTGTCAGTCCTTTCATC	8 GA	254	50.4
				R-AAAGGTTAAGGGTCCTGCTAGC			52.9
AC235795	C03HBa0036B17	ju004	3	F-TCGATGTCATTACTCACGTTCC	5 CA	238, 242	51.7
				R-GATACCAAAACGCAGCAAGTTG			54.1
AC235804	C03HBa0224P23	ju006	10 *	F-CATTTCATGAAAGGGGAATTCTAG	10 TG	201, 277	53.1
				R-ACATTTCGTGTTAGCTGGGTTC			52.6
AC238438	C03HBa0031F10	ju 007	3	F-GAGTTTGATAAAGCAAAAGGC	6 AG	163, 182	48.2
				R-AACAGAACCCGAGTTTGGAC			50.5
AC238439	C03HBa0031P17	ju008	3	F-CAATTATTAGACAGCCAACCAAG	5 AAT	264	50.5
				R-GGCATTTATTTGGTCAGAAAGC			52.5
AC238450	C03HBa0114P24	ju010	3	F-TACCCTTTCGTTTACCCAAATTG	11 AT	282	54.1
				R-AATTGACCGATTTTCCCTTCTC			53.1
AC238451	C03HBa0121O11	ju011	3	F-GTGAAATGATGTTTCCTCTGACAAG	5 AAC	246, 253	53.7
				R-CTTTCGACATCCTTTTGACTCG			53.2
AC238457	C03HBa0166B15	ju014	3	F-CGGCAATGTAAGAGTTGAGCTC	6 GA	243	53.4
				R-ATCATCCCAAGCGTCAAAATAG			52.7
AC238459	C03HBa0176B05	ju015	3	F-ACTCTTCATCCGTTGTACAATTC	6 TTC	264, 276	49.5
				R-TTCACTCGGATGATTGTAATCG			51.9
AC238462	C03HBa0203H10	ju017	3	F-GATTTTATTGGGTGTCTGTTGTC	5 TGT	248	49.8
				R-AGGGAGAAAAGATGAACAGTATC			48.1
AC238468b	C12HBa0270F24	ju022	12 **	F-ATGGATTTACTGTAACAGTGTGAAC	6 TTC	293	49
				R-GTCCAAATTAATAACAGATCCATAG			48.4
AC238468c	C12HBa0270F24	ju023	12 **	F-AATTATTCGTAAGTTTCCGTCTGTC	25 AT	308, 320	52.2
				R-CCTTTATGAATGACCAAAAGCTAC			51.3
AC238560	C03HBa0030A11	ju026	3	F-AATCAATATCATCGCTTCACTG	19 TA	246, 292	48.9
				R-ATGTTGTGGTATTATTGACTTATGAG			48.7
AC238561	C03HBa0031M05	ju027	3	F-ATGCTTAAGGTCTCCAAACACC	5 CAA	250	51.8
				R-CTCTCTACTTTTGGGATTACGC			49.7
EU124730	C03HBa0001E24	ju029	3	F-TGCTGTACATACTGCATAAATGG	7 TG	350	50.1
				R-AACCTGCTGAATTAACTTGTAGTG			49.5
EU124737	C03HBa0054O21	ju035	3	F-GTTATATAGAAAGACAAGGTAGAAGGTC	25 AT	288, 293	49.7
				R-GGTAGACTTTTTATGTGTTGTTGC			49.7
EU124739	C03HBa0233O20	ju037	3	F-AAAATTGTTGGTCAACATGGTG	7 TAT	241, 246	51.6
				R-TTATCTCCTTTCCCTTTCATTC			49
EU124741	C04HBa0318C22	ju039	4	F-GATGGTGTCATAGATCTAGCCTTAG	6 TTAA	355, 421	50.4
				R-TGGGGAATTATGTAGTGTTGAG			48.7
EU124742	C03HBa0323D22	ju040	3	F-GCGATCCTGTTTGAGAAGAAGG	5 CA	340, 345	54.6
				R-ATGAACAAATGCTTAAGAGGGG			52
EU124743	C03HBa0007J09	ju041	3	F-TTCCAAAAACACTTACGAAAGTTAG	26 AT	292, 316, 330	51.4
				R-CATGTAAGTCAAAAGAATGGAGG			50.2

* The original clones was assigned to chromosome number 3 (clone number C03HBa0224P23). ** The original clones was assigned to chromosome number 3 (clone number C03HBa0270F24).

**Table 3 biotech-11-00034-t003:** SSR locus name, number of alleles, (Ho), (He) observed and expected heterozygosities, polymorphic information content (PIC), and discrimination power (PD) values for 22 developed polymorphic microsatellite loci in a sample of 29 tomato landraces.

Locus	No. of Alleles	Ho	He	PIC	PD
ju003	1	0	0	0	0.32
ju004	2	0.24	0.41	0.32	0.72
ju006	2	0.59	0.49	0.36	0.64
ju007	2	0.81	0.5	0.37	0.34
ju008	1	0	0	0	0.61
ju010	1	0	0	0	0.73
ju011	2	0.31	0.27	0.23	0.68
ju014	1	0	0	0	0.52
ju015	2	0.32	0.27	0.23	0.75
ju017	1	0	0	0	0.37
ju022	1	0	0	0	0.48
ju023	2	0	0.42	0.32	0.96
ju026	2	0.71	0.54	0.38	0.96
ju027	1	0	0	0	0.73
ju029	1	0	0	0	0.52
ju035	2	0	0.42	0.32	0.94
ju037	2	0	0.5	0.37	0.92
ju039	2	0.48	0.37	0.3	0.72
ju040	2	0.09	0.09	0.08	0.83
ju041	3	0.26	0.51	0.38	0.91
Total	33	-	-	-	
Mean	1.65	0.19	0.24	0.18	0.68
Max	3	0.81	0.54	0.38	0.96
Min	1	0	0	0	0.32

**Table 4 biotech-11-00034-t004:** Potential SSR markers and nearby genes.

Locus	Nearby Gene	Functions	Reference
ju014, ju015, ju035	Serine/threonine-protein kinase	Central processor unit (cpu): accepting input information from receptors that sense environmental conditions, phytohormones, and other external factors, and converting it into appropriate outputs, such as changes in metabolism, gene expression, and cell growth and division	[40]
ju017	Serine carboxypeptidase	Stress response, growth, development, and pathogen defense	[41]
ju023	Pirin	Role in seed germination and transcription of a light- and ABA-regulated gene under specific conditions in Arabidopsis thaliana	[42]
ju010. ju029	F-box family protein	Plant hormonal signal transduction, floral development, secondary metabolism, senescence, circadian rhythms, and responses to both biotic and abiotic stresses	[43]
ju014, ju040	Polygalacturonase	Major role in cell wall degradation and fruit softening	[44]

## Data Availability

Not applicable.

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
