# Peer review of "Development of SSR Markers Linked to Stress Responsive Genes along Tomato Chromosome 3 (Solanum lycopersicum L.)"

_biotech, 2022, doi:10.3390/biotech11030034_

Round 1

Reviewer 1 Report

The object of the submitted manuscript is an important agricultural crop - tomato. Therefore, all work. related to improving the growing conditions of this valuable crop, increasing its yield, are of value. The main remark concerns the formulation of the problem and some conclusions.

There is no clear indication: 1) different tomato varieties grown in different geographical areas were used or 2) the same tomato varieties grown in different different geographical areas. If 1), then how can we talk about the influence of a geographical zone if it is a different variety of tomato. If 2), then it is necessary to give a description of the geographical area, soil, etc. Further, all varieties of tomatoes were grown under the same laboratory conditions and, therefore, only the characteristics of the variety can be discussed.

In Annotations. It is not clear on the basis of the content of what? the choice was made.

-The introduction section mainly describes the importance of the tomato as an agricultural crop. Expand the application SSR markers at the abiotic stress  in agriculture.

- in the caption to Fig. 2 - error

- unscramble SRAP marker

Author Response

Does the introduction provide sufficient background and include all relevant references?

Was improved as requested

Are all the cited references relevant to the research?

Recent relevant articles were added and cited

Is the research design appropriate?

We thank the reviewer for this comment and it will be taken in consideration in future work.

Are the methods adequately described?

Was improved as requested

Are the results clearly presented?

It was improved as requested

Are the conclusions supported by the results?

It was improved as requested

Comments and Suggestions for Authors

The object of the submitted manuscript is an important agricultural crop - tomato. Therefore, all work. related to improving the growing conditions of this valuable crop, increasing its yield, are of value.

Thank you for your positive remark

 The main remark concerns the formulation of the problem and some conclusions.

The formulation was improved as requested

And the conclusions were improved

There is no clear indication:

1) different tomato varieties grown in different geographical areas were used or 2) the same tomato varieties grown in different different geographical areas. If 1), then how can we talk about the influence of a geographical zone if it is a different variety of tomato. If 2), then it is necessary to give a description of the geographical area, soil, etc.

The first one (1) is the correct statement. The utilized landraces were collected from 13 different geographical regions (mainly from north and south of Jordan), please see the following cited reference for details " Brake M.H.; Al-Gharaibeh, M.A.; Hamasha, H.R.; Al-Sakarneh, N.S.; Alshomali, I.A.; Migdadi, H.M.; Qaryouti, M.M.; Haddad, N.J. Assessment of genetic variability among Jordanian tomato landrace using inter-simple sequence repeats markers. Jordan Journal of biological science 2021, 14(1), 91-95."

The purpose of using landraces (genetically diverse plant material)  was to validate the developed SSR markers.

Further, all varieties of tomatoes were grown under the same laboratory conditions and, therefore, only the characteristics of the variety can be discussed.

This is true, however, we utilized these landraces for marker validation.

In Annotations. It is not clear on the basis of the content of what? the choice was made.

The markers were annotated based on the affiliated universities (Jerash University and university of Jordan.

-The introduction section mainly describes the importance of the tomato as an agricultural crop. Expand the application SSR markers at the abiotic stress  in agriculture.

An important review paper was cited and added as requested in the introduction " Younis, A.; Ramzan, F.; Ramzan, Y.; Zulfiqar, F.; Ahsan, M.; Lim, K.B. Molecular markers improve abiotic stress tolerance in crops: a review. Plants 2020, 9(10), 1374."

- in the caption to Fig. 2 - error

Corrected

- unscramble SRAP marker

full name was added

Reviewer 2 Report

Manuscript Development of SSR markers linked to stress responsive genes along tomato chromosome 3 (Solanum lycopersicum L.) by Mohammad Brake, Lana Al-Qadumii, Hassan Hamasha, Hussein Migdadi, Abi Awad, Nizar Haddad,

Monther Sadder considers a possible promising approach to evaluate different tomato genotypes for the presence of markers presumably associated with resistance to abiotic stressors.

Although at present this technology is at the initial stage of development and use in selection evaluation and may later be changed and even challenged, attempts to use this technology are currently promising.

There are a few comments on the design of the tables - they are not made according to the rules and are poorly readable, perhaps this material should be enclosed in a supplement.

It would also be reasonable to accompany the conclusion with arguments about the prospects for using the obtained data in breeding. The work is based on a variety of materials, contains new data and can be accepted for publication in its present form.

Author Response

Does the introduction provide sufficient background and include all relevant references?

Was improved as requested

Are all the cited references relevant to the research?

Recent relevant articles were added and cited

Are the results clearly presented?

It was improved as requested

Are the conclusions supported by the results?

It was improved as requested

Comments and Suggestions for Authors

Manuscript Development of SSR markers linked to stress responsive genes along tomato chromosome 3 (Solanum lycopersicum L.) by Mohammad Brake, Lana Al-Qadumii, Hassan Hamasha, Hussein Migdadi, Abi Awad, Nizar Haddad, Monther Sadder considers a possible promising approach to evaluate different tomato genotypes for the presence of markers presumably associated with resistance to abiotic stressors.

We thank the reviewer for this comment

Although at present this technology is at the initial stage of development and use in selection evaluation and may later be changed and even challenged, attempts to use this technology are currently promising.

We thank the reviewer for this comment

There are a few comments on the design of the tables - they are not made according to the rules and are poorly readable, perhaps this material should be enclosed in a supplement. We thank the reviewer for this comment, the table 3 was allocated to supplementary material. It would also be reasonable to accompany the conclusion with arguments about the prospects for using the obtained data in breeding. The work is based on a variety of materials, contains new data and can be accepted for publication in its present form.

Were modified as requested

Reviewer 3 Report

This study was aimed to develop novel SSR markers in tomato. Several BAC clones along  chromosome 3 in tomato were selected based on their content.  this manuscripts is meanful and attract more reader .especially for tomato breeders.

all the tomato materials is numbers , I do not think the number is Accession Number of  tomato. please describe detail where is material from? and add references in right place.

1 question:

why your research only focus on Chromosome 3?  all the stress related genes in Chromosome 3?   

Author Response

Are all the cited references relevant to the research?

Recent relevant articles were added and cited

Are the conclusions supported by the results?

It was improved as requested

Comments and Suggestions for Authors

This study was aimed to develop novel SSR markers in tomato. Several BAC clones along chromosome 3 in tomato were selected based on their content. this manuscripts is meanful and attract more reader .especially for tomato breeders.

We thank the reviewer for this comment

all the tomato materials is numbers , I do not think the number is Accession Number of tomato. please describe detail where is material from? and add references in right place.

Please refer to 2.1 Plant materials section. These number are accession numbers (Table 1), which is used by the seed provider (National Agricultural Research Center (NARC)) as a code.

1 question:

why your research only focus on Chromosome 3? all the stress related genes in Chromosome 3?

Different published articles found that chromosome 3 in tomato contain both biotic and abiotic related genes (see references 7 and 9.